# Loneliness in Intimate Relationships Scale (LIRS): Development and Validation

**DOI:** 10.3390/ijerph191912970

**Published:** 2022-10-10

**Authors:** Ami Rokach, Ami Sha’ked, Elisheva Ben-Artzi

**Affiliations:** 1Department of Psychology, York University, Toronto, ON M3J 1P3, Canada; 2Academic Center for Law & Business, Ramat Gan 5224213, Israel

**Keywords:** loneliness, intimacy, intimate relationships, marriage, assessment

## Abstract

Intimate relationships have been shown to be loneliness positively related to self-esteem. Happiness and well-being and have also been regarded as a buffer against loneliness. Nevertheless, substantive research indicates that intimate relationships and marriage can produce or result in loneliness and thus seriously affect the person’s physical, emotional and spiritual well-being. Loneliness in intimate relationships may damage the relationship if it goes on, and thus, this newly developed scale has been introduced to aid clinicians and researchers in discovering loneliness in an intimate union so it can be addressed before it negatively affects the union. Since none of the measures of loneliness tap loneliness as experienced in intimate relationships, a new rating scale, the Loneliness in Intimate Relationships Scale (LIRS), was developed and tested psychometrically. The generation of items followed a qualitative approach based on a semi-structured questionnaire administered to 108 volunteers from the general Israeli population, theoretical and empirical literature, and assessments of expert psychologists. In a second study (N = 215), a self-report scale assessing loneliness in intimate relationships was developed. This was followed by psychometric and construct validity evaluations with a new sample of 306 participants. Analyses revealed that loneliness in intimate relationships is experienced mainly in terms of three aspects: detachment, hurt, and guilt. Exploratory and confirmatory factor analyses and validity tests indicate that the final 14-item Loneliness in Intimate Relationship Scale is a well-structured, reliable, and valid scale tapping emotional, behavioral, and cognitive manifestations of loneliness in intimate relationships.

## 1. Loneliness in Intimate Relationships Scale (LIRS): Development and Validation

Establishing and maintaining close intimate relationships with a significant other has been recognized as a fundamental human motivation [1,2]. Similarly, marriage is perceived as the most intimate officially sanctioned adult bonding, serving as a primary source of affection, love, support [3,4] and physical and emotional well-being [5].

In the US, approximately sixty percent of adults (60% of males and 57% of females) aged 18 and over are married, while overall, 72% have experienced marriage and may thus be married, divorced, or widowed. Marriage, or a long-term intimate commitment, is central in Western culture, and although they presently have smaller chances of succeeding, they are still, by and large, the preferred lifestyles adult [4]. Strong [4] suggested that intimate relationships buffer against loneliness, positively affect our self-esteem, and are shown to be related to happiness, contentment, and a sense of well-being [6]. In close relationships, intimacy has been rated as more important for relationship satisfaction than autonomy, individuality, freedom, agreement, or sexual satisfaction [7].

This, however, does not prevent the married from experiencing loneliness. For example, Tornstam [8] found that 40% of married people in Sweden experienced more loneliness than unmarried people. It may intuitively seem paradoxical that a person can be both married and lonely. However, when marriages lose their vitality, spouses become prone to loneliness [9]. There is a tendency to believe that marriage and intimate relations tend to fend off loneliness because a companion is always around. However, Barbour [10] found that 20% of wives and 24% of husbands are significantly lonely, and as loneliness increases in the marriage or intimate relationship, so does depression.

Loneliness in marriage may be especially distressing because it is inconsistent with expectations about marriage and may have a significant effect on one’s physical, emotional and spiritual well-being [11,12]. As indicated by Fincham and Rogge [13], there are two approaches that examine the construct of relationship quality. One focuses on the relationship, or the interpersonal exchange between the couple, namely their interaction, the manner in which they resolve conflicts, and their communication patterns. The other, the intrapersonal approach, which the present study took, focuses on subjective judgments of each partner and her or his evaluation of the marriage or intimate relationship.

Helping couples deal with conflicts, problematic issues, or dissatisfaction is what marital and couple therapists commonly do. However, in order to be effective, therapy needs to be clear as to what to focus on and address the specific aspects of the relationship that need the therapist’s help and attention. There are a variety of assessment tools geared to examine couple relationships and problematic issues, including the Locke–Wallace Marital Adjustment Scale [14] the Quality of Relationship Inventory [15], the Brief Romantic Relationship Interaction Coding Scheme [16], the Kansas Marital Satisfaction Scale [17], and the Retired Spousal Intrusion Scale [18]. However, although important, valid and reliable instruments have been developed for assessing different aspects of relationships, none address the issue of loneliness, which can be highly disruptive and damaging to an intimate relation.

We, therefore, developed a new scale, the Loneliness in Intimate Relationships Scale [LIRS], which aims to fulfill this need. It is expected that there may be a moderate association between the present general loneliness scales and the one described herein since the one that we developed addresses relational issues that were not addressed in the existing loneliness questionnaires. The present manuscript does not check this assumption, and that will be addressed by our planned future research project. The present paper describes research aimed at developing a new scale for assessing loneliness in relationships for adults and testing its basic psychometric properties regarding reliability and validity. The development of the Loneliness in Intimate Relationships Scale followed the standard principles of a non-reactive methodology for questionnaire construction described in the *Standards for Educational and Psychological Testing* (APA, 2014). Such methodology ensures that the scale covers all aspects of the construct and meets the highest validity and reliability criteria. Table 1 presents the stages in the development of the Loneliness in Intimate Relationships Scale.

## 2. Study 1

Generating items for the loneliness in intimate relationships scale.

Study 1 was designed to generate a large pool of items relevant to loneliness in intimate relationships, which would serve as the pool for developing the final scale. In the present study, the collection of items was based on both naive participants’ descriptions and professional literature and therapists’ input evaluations based on clinical practice relevant to loneliness in intimate relationships. Study 1 was therefore carried out as a qualitative exploration aimed at assessing items and domains of loneliness in relationships.

### 2.1. Method

#### 2.1.1. Participants

A total of 108 participants from the general Israeli population (70 women) volunteered to participate in the study. The participants represented a broad range of demographic characteristics and were solicited from all walks of life. Age ranged between 19 and 59 years (M = 36.09, SD = 10.71). Education ranged between 10 and 18 years (M = 14.00, SD = 2.17). Seventy-nine (73%) of the participants were in a steady relationship at the time of the study. The study was approved by the Institutional Review Board of the Center for Academic Studies.

#### 2.1.2. Semi-Structured Questionnaire

A semi-structured questionnaire was administered, in which the participants were asked to freely describe their reflections and feelings during an experience in which they felt lonely during a close/intimate relationship. The participants received the following instructions, “We are attempting to understand loneliness in intimate relationships by investigating how it is experienced. We would appreciate your reflections on an experience of loneliness in an intimate relationship. Please refer to a specific period or situation in which you felt lonely in an intimate relationship and describe what you felt or thought through the following five guiding aspects, as listed below. It is important that you to realize that there are no “right” or “wrong” responses to these aspects/questions. People are different, and we are interested in your personal experience.”

The participants were then asked to freely refer to the following aspects of the situation: (1) “Describe, in a number of sentences, the situation/period in which you felt lonely”; (2) “Describe the thoughts you had about yourself and the relationship”; (3) “Describe the feelings you had about yourself and the relationship”; (4) “What did you want to happen to change the feeling of loneliness?”; and (5) “How did you deal with the situation?” The participants were not given a time limit for answering the questionnaire.

### 2.2. Results and Discussion

#### 2.2.1. Listing and Categorization of the Data

A qualitative content analysis was performed on the 108 transcripts by two psychology students after having been trained by the researchers. Data were analyzed according to the following stages: (1) reduction of descriptions to precise terms; (2) identification of separate single-content statements; (3) removal of redundant items; (4) initial categorization of items into clusters. The descriptions reported by the participants yielded a wide range of statements. In the first stage of the analysis, two psychology students read all 108 protocols and recorded all the statements. Of the 108 transcripts, 533 separate single-content statements (verbatim) were identified (e.g., “I felt anger,” “I thought our relationship was over”). Next, similarly worded statements were given a common label. For example, “I felt lonely” and “I sensed loneliness” were grouped together. Decisions on item grouping were carried out based on the two judges’ full agreement. A list of 156 different items that were mentioned by at least one of the 108 participants was obtained.

The three authors then grouped the listed items into non-overlapping categories that they proposed. Items assessing similar issues were gathered to make up a subscale (category). The authors performed the categorization task separately. Inter-judge reliabilities were computed using a conservative method of percent agreement between judges for each category separately (i.e., the percentage of items that two judges grouped in the same category). The inter-judge agreement on all categories ranged between 0.80 and 0.90. Items for which there was no judge agreement were removed from the list. This stage resulted in a 138-item list.

#### 2.2.2. Analysis of Existing Relevant Scales and Clinical Practice

Finally, we made a thorough analysis of existing relevant scales, as follows: Social and Emotional Loneliness Measure [19], de Jong Gierveld Loneliness Scale [20], Differential Loneliness Scale [21], Loneliness-Deprivation Scale [22], Loneliness and Social Dissatisfaction Questionnaire—Modified [23], Loneliness Questionnaire [24], Social and Emotional Loneliness Scale for Adults [25], Revised UCLA Loneliness Scale [26], Social and Emotional Loneliness Scale for Adults [27], and the Social and Emotional Loneliness Scale for Adults—Short Version [28]. Appraisals derived from the above-mentioned scales and clinical practice experience yielded an additional 28 items relevant to loneliness in intimate relationships. These items were added to the item list derived from the semi-structured questionnaires’ content analysis, such that when an item was identified as belonging to one of the categories but did not appear in it, the judges added it to that category. When an item could not be grouped into any of the proposed categories, the judges created a new category for it. This procedure was carried out separately by three judges: two of the authors (E.B. & A.R.) and an additional psychology student. Inter-judge reliabilities were computed, yielding an inter-judge agreement on all categories ranging between 0.81 and 0.89. Items for which there was no full judge agreement were removed from the list. This procedure resulted in a list of 123 items grouped into 29 content categories.

Next, each item of the 123-item list was evaluated by two psychology students who worked separately regarding its clarity and comprehensibility. Items that did not meet these criteria were removed from the list or were rephrased based on an agreement by the three authors. This procedure resulted in a 66-item list which comprised the first version of the Loneliness in Intimate Relationship Scale. Table 2 presents the 29-category list derived from the content analysis of the semi-structured questionnaires, existing relevant scales and clinical practice. Each category consisted of 2–3 relevant items (e.g., “I felt sadness,” “I felt that our love is fading”).

## 3. Study 2

Study 2 was designed to develop a self-report scale assessing loneliness in intimate relationships based on the results of the content analysis described in Study 1.

### 3.1. Method

#### 3.1.1. Participants

A total of 215 participants from the general Israeli population (181 women) volunteered to participate in the study. Age ranged between 20 and 64 years (M = 33.95, SD = 10.49). Education ranged between 8 and 16 years (M = 12.10, SD = 0.96). Overall, 148 (69%) of the participants were in a steady relationship at the time of the study, while 65 (30%) were not. Two participants did not indicate their relationship status. A total of 43 (20%) participants reported feeling lonely at the time of the study, and 69 (32%) reported that when they experienced loneliness, it was “on a more or less continuous basis.” The study was approved by the Institutional Review Board of the Center for Academic Studies.

#### 3.1.2. Loneliness in Intimate Relationships Scale—Version 1

The first version of the Loneliness in Intimate Relationships Scale (LIRS) consisted of the 66 items derived from Study 1 (see Appendix A). The participants were given the following instructions: “We are attempting to understand loneliness in intimate relationships by investigating how it is experienced. We would appreciate your reflections on an experience in which you felt lonely during a close/intimate relationship. Please read each of the following statements and decide how much it describes what you felt or thought during a specific period or situation in which you felt lonely in a close relationship. Please respond according to the following 6-point scale ranging from 1 = totally not describes my experience to 6 = totally describes my experience. It is important for you to realize that there are no “right” or “wrong” answers to these questions. We are interested in how you felt. Please note that in this questionnaire, partner refers to a romantic ‘partner’ of either gender.”

### 3.2. Results and Discussion

#### Exploratory Factor Analysis

The analysis was based on the responses of 215 participants, representing an adequate participants-per-item ratio (1:3.3) [29]. The ratings of the 66 items of the first version of the Loneliness in Intimate Relationship Scale were subjected to a principal components factor analysis with oblique factor rotation. Because rotated factors were only modestly correlated ($\bar{r}$ = 0.39), we reanalyzed the data using a varimax factor rotation. The number of factors to extract was determined by parallel analysis, which has been shown to provide more accurate estimates of the number of factors to retain than Kaiser’s criterion of eigenvalues > 1 [30]. To establish the level for meaningful eigenvalues we conducted a principal-components factor analysis on random data (“Monte Carlo”) generated from the raw data. We used O’Connor’s SPSS software [31] to generate the random data set and to compute eigenvalues (and 95% CIs) on the random data set. Each parallel data set is based on column-wise random shufflings of the values in the raw data matrix using Castellan’s (1992, BRMIC, 24, 72–77) algorithm. The distributions of the original raw variables are exactly preserved in the shuffled versions used in the parallel analyses. Permutations of the raw data set are thus. We computed 10,000 datasets, which is considered more than sufficient. Parallel analysis of the 66 items based on the mean eigenvalues and 95th eigenvalue obtained from random data indicated a four-factor solution accounting for 55.4% of the items’ variance, with the first factor accounting for 41.6% of the total variance. However, several items had poor factor loadings on all the factors (L < 0.40) or had high factor loadings on more than one factor (L ≥ 0.40) and were therefore removed from the list. The remaining 14 items were subjected to a principal components factor analysis with oblique factor rotation. Because rotated factors were only modestly correlated ($\bar{r}$ = 0.35), we reanalyzed the data using a varimax factor rotation. Parallel analysis of the 14 items based on the mean eigenvalues and 95th eigenvalue obtained from random data indicated a three-factor solution, accounting for 61.8% of the items’ variance, with the first factor accounting for 41% of the total variance. Factor 1 includes six items: 5, 8, 10, 11, 12, 13; factor 2 includes four items: 1, 2, 3, 4; and factor 3 also includes four items: 6, 7, 9, 14. Table 3 presents item loadings and factors’ statistics. Inspection of item loadings in Table 3 indicates that all items have distinct loadings on the three factors. Analysis of item content suggests that factor 1 represents thoughts and feelings concerning detachment and separation, factor 2 represents feelings of hurt and pain, and factor 3 represents a sense of guilt and responsibility. The final 14-item version appears in Appendix A.

## 4. Study 3

Study 3 was designed to validate the structure pattern of the 14-item version of the LIRS that emerged in study 2 and to analyze its psychometric properties.

### 4.1. Method

#### 4.1.1. Participants

A group of 306 participants from the general Israeli population (158 women) volunteered to participate in the study. They were recruited from all walks of life. Age ranged between 18 and 54 years (M = 26.99, SD = 6.95). Education ranged between 8 and 14 years (M = 11.99, SD = 0.44). A total of 135 (44%) of the participants were in a steady relationship at the time of the study, 169 (56%) were not. Two participants did not indicate their relationship status. Overall, 69 (23%) participants reported feeling lonely at the time of the study, and 87 (31%) reported that when they experienced loneliness, it was “on a more or less continuous basis.” The study was approved by the Institutional Review Board of the Center for Academic Studies.

#### 4.1.2. Loneliness in Intimate Relationships Scale—Version 2 (14 Items)

The 14-item LIRS version was used. The participants were given the instructions described in Study 2.

### 4.2. Results and Discussion

#### 4.2.1. Exploratory Factor Analysis

In order to further ensure the structure validity of the LIRS, the goodness-of-fit to the data of the three-factor solution extracted from the exploratory factor analysis was compared to an equivalent random model using a structural equation modeling approach (AMOS, SPSS 21.0). The sample size (N = 306) was adequate for a confirmatory analysis (Kline, 2011). The equivalent random model was comprised of items 1, 5, 6, 7, 11, and 13 in factor 1, items 2, 3, 9, and 12 in factor 2, and items 4, 8, 10, and 14 in factor 3. The evaluated goodness-of-fit indicators included the overall χ^2^. The evaluated descriptive indexes were the incremental fit index (IFI), the comparative fit index (CFI), and the root-mean-square error of approximation (RMSA), being complementary descriptive indexes (Schumacker & Lomax, 2012). The results of the confirmatory analyses comparing the goodness-of-fit to the data of the original and random three-factor models for the final 14-item version of the LIRS are presented in Table 4.

As shown in Table 4, the original three-factor model fit the data well. Although the χ^2^ was significant (which is expected with large samples), all the descriptive indexes indicated a satisfactory goodness-of-fit. Moreover, the χ^2^ difference between the original and random models was significant, and all descriptive indexes of the random model indicated a poor fit to the data for the random model. Standardized factor coefficients are presented in Figure 1. Finally, as indicated in Figure 1, the three factors were moderately related.

Taken as a whole, the exploratory and confirmatory factor analyses indicate that the final 14-item version of the Loneliness in Intimate Relationship Scale (LIRS) is a well-structured and reliable scale depicting emotional, behavioral, and cognitive reactions and coping with loneliness in intimate relationships.

#### 4.2.2. Confirmatory Factor Analysis

In order to further ensure the structure validity of the LIRS, the goodness-of-fit to the data of the three-factor solution extracted from the exploratory factor analysis was compared to an equivalent random model using a structural equation modeling approach (AMOS, SPSS 21.0). The sample size (N = 306) was adequate for a confirmatory analysis (Kline, 2011). The equivalent random model was comprised of items 1, 5, 6, 7, 11, and 13 in factor 1, items 2, 3, 9, and 12 in factor 2, and items 4, 8, 10, and 14 in factor 3. The evaluated goodness-of-fit indicators included the overall χ^2^. The evaluated descriptive indexes were the incremental fit index (IFI), the comparative fit index (CFI), and the root-mean-square error of approximation (RMSA), being complementary descriptive indexes (Schumacker & Lomax, 2012). The results of the confirmatory analyses comparing the goodness-of-fit to the data of the original and random three-factor models for the final 14-item version of the LIRS are presented in Table 4.

As shown in Table 4, the original three-factor model fit the data well. Although the χ^2^ was significant (which is expected with large samples), all the descriptive indexes indicated a satisfactory goodness-of-fit. Moreover, the χ^2^ difference between the original and random models was significant, and all descriptive indexes of the random model indicated a poor fit to the data for the random model. Standardized factor coefficients are presented in Figure 1.

Taken as a whole, the exploratory and confirmatory factor analyses indicate that the final 14-item version of the Loneliness in Intimate Relationship Scale (LIRS) is a well-structured and reliable scale depicting emotional, behavioral, and cognitive reactions and coping with loneliness in intimate relationships.

## 5. Study 4

The purpose of this study was to assess the construct validity of the LIRS. Specifically, we performed two sets of concurrent validity tests. We tested the relationships between the subscales of the LIRS and other scales tapping close but different constructs, namely general loneliness, social loneliness, interpersonal needs, and depression. All these concepts were expected to be moderately positively associated with all LIRS subscales. We also compared the levels of loneliness in intimate relationships as established by the LIRS subscales for individuals who described their experience of loneliness in intimate relationships as continuous with those who described it as a specific state-related event. It was predicted that levels of loneliness in intimate relationships would be elevated for the former group. All of the above evidence would provide further support for the psychometric strength of the LIRS.

### 5.1. Method

#### 5.1.1. Participants

The 306 individuals who participated in study 3 took part in the present study.

#### 5.1.2. Measures

The 14-item version of the Loneliness in Intimate Relationships Scale (LIRS) developed in studies 1–3 was used. The participants were given the instructions described in study 3. This final version of the LIRS was comprised of three subscales: detachment (6 items), hurt (4 items), and guilt (4 items). The score for each subscale is computed by averaging the ratings of the relevant items, with higher scores reflecting greater feelings of detachment, hurt, and guilt. Study 4 revealed satisfactory internal reliabilities for all three subscales (α = 0.86, 0.87, 0.64 for the detachment, hurt, and guilt subscales, respectively.

The Revised UCLA Loneliness Scale (UCLA-R) [32] is the most widely used self-report measure of loneliness for both adolescents and adults. The scale consists of 20 items, of which 10 are positive (e.g., “There are people I can talk to”) and 10 are negative (e.g., “People are around me but not with me”). Respondents are asked to indicate how often (1 = never, 2 = rarely, 3 = sometimes, or 4 = often) they feel the way described in each item. Positive items (1, 4, 5, 6, 9, 10, 15, 16, 19 and 20) were reverse-coded prior to analysis. The total score is the sum of all 20 items, with higher scores reflecting greater feelings of loneliness. The UCLA-R scale has been found to have high internal consistency, with α = 96 and test-retest reliability = 0.94 [26,33]. In the present study, the UCLA-R also had a high internal consistency (α = 0.91).

The Social and Emotional Loneliness Scale for adults—short version (SELSA-S) is a 15-item questionnaire that measures loneliness as a multidimensional construct. The SELSA-S has three 5-item subscales: romantic loneliness, family loneliness, and social loneliness. In the present study, we used only the social loneliness subscale (e.g., “I feel part of a group of friends”). Items are rated on a 7-point Likert scale that ranges from 1 (strongly disagree) to 7 (strongly agree). Items 1, 2 and 4 were reverse-coded prior to analysis. The total score is the mean of all 5 items, with higher scores reflecting greater feelings of social loneliness. The SELSA-S has been shown to be a reliable and valid measure of adult loneliness [28]). In the present study, the social loneliness subscale had high internal reliability (α = 0.82).

The Interpersonal Needs Questionnaire (INQ) [34,35] is a 15-item self-report measure of interpersonal needs tapping perceived burdensomeness (6 items) and thwarted belongingness (9 items). Respondents indicate the degree to which statements are true for them on a 7-point Likert-type scale that ranges from 1 (“not true for me at all”) to 7 (“very true for me”). Items 7, 8, 10, 13, 14 and 15 were reverse-coded prior to analysis. The total scores are the mean of all the items in the relevant subscales, with higher scores reflecting greater feelings of perceived burdensomeness and thwarted belongingness. The INQ has shown evidence of high levels of validity (Van Orden et al., 2012). Internal consistencies of the INQ scales in the current study were high (INQ-TB α = 0.92, INQ-PB α = 0.74).

The Beck Depression Inventory (BDI) [36] is a questionnaire consisting of 21 groups of statements referring to different aspects of depression. Respondents are asked to endorse statements characterizing how they have been feeling throughout the past two weeks. The maximum total score for all 21 items is 63. The BDI has shown evidence of high levels of validity and reliability [36,37].

All the scales were administered in a counter-balanced order in two forms. No order effect was found.

### 5.2. Results and Discussion

The means and standard deviations for the different measures are presented in Table 5. Pearson correlations among the LIRS subscales (see Table 6) suggest that detachment is highly associated with hurt (r = 0.63), whereas guilt has lower associations with both detachment (r = 0.28) and hurt (r = 0.34).

The relationships between the LIRS three subscales and loneliness and depression measures were computed in order to assess the validity of the LIRS (see Table 6). Bonferroni’s formula was used to adjust the α level from 0.05 to 0.05 divided by fifteen, or 0.003. Only values of *p* ≤ 0.003 were considered to be significant. It was hypothesized that all three subscales of the LIRS would be positively but moderately associated with all other measures.

As indicated in Table 6, most of the correlations between the LIRS subscales and the criteria variables were in the expected positive direction, indicating that greater loneliness in intimate relationships is related to higher levels of reported general and social loneliness and greater interpersonal needs and depression. The non-significant associations between the detachment and hurt subscales of the LIRS and the perceived burdensomeness subscale of the INQ provide support for the discriminant validity of the LIRS, which refers to the distinctiveness of different constructs [38]. These results validate the LIRS as tapping loneliness and social detachment tendencies, yet measuring new and unique aspects of loneliness, namely loneliness in intimate relationships, which are not addressed by any other measure.

To further validate the LIRS as a measure of loneliness in intimate relationships, we compared the patterns of its subscales between individuals who reported that when they experienced loneliness, it was on a more or less continuous basis, with those reporting that it was related to a specific occasion. The results are presented in Table 7.

A multivariate analysis of variance (MANOVA) conducted on the three LIRS subscales means with the type of loneliness (continuous/specific) as IV revealed a significant multivariate effect of type of loneliness, *F* (3, 278) = 9.09, *p* < 0.001. As expected, univariate analyses (see Table 7) revealed that the level of loneliness in intimate relationships is greater for individuals who experienced loneliness in a relationship on a continuous, not state-related basis, compared with individuals whose loneliness experience was more specific and state-related. This pattern emerged for all three LIRS subscales.

Finally, despite not having preliminary hypotheses regarding gender differences in LIRS scales, we wished to explore them. The results are presented in Table 8. A multivariate analysis of variance (MANOVA) conducted on the three LIRS subscales means with gender as IV revealed a significant multivariate effect of gender, *F* (3, 299) = 9.14, *p* < 0.001. Univariate analyses revealed that women experience significantly higher levels of detachment and hurt, while men experience significantly higher levels of guilt.

Taken as a whole, the results of Study 4 provide support for the validity properties of the LIRS.

## 6. Conclusions

Various pitfalls, stumbling blocks, and hazards, such as hurt feelings, ostracism, jealousy, lying, and betrayal, may harm emotional relations and feelings of love [39]. We want and need to be loved by our intimate partners and hope that our relational value—the degree to which our partner considers our intimate relationship valuable, important, and close—is as high as we perceive it to be. It is painful to discover that our partner may perceive it as lower than we would like it to be perceived. A dissonance is thus created between what we envision it to be and what our partner apparently sees in it. Should relational devaluation occur, and we are no longer thought of as positively as before, we experience pain, anger, hurt, and loneliness [40,41] described ideal romantic love as a situation where the one I love loves me back since “Out of all the people she could love, she chooses to love me. That suggests that the reason why she loves me should be to do with the things that set me apart from others” (p. 163). When those feelings seem to change, when one stops feeling so valued and special to their partner, it may lead to their feeling neglected, unappreciated, and thus lonely. When one is not as important to one’s lover as one knows one was in the past, it ushers in loneliness, sadness, and longing to recreate what was, what made one feel so good and special in the first place.

The newly created scale is expected to be valuable, not only for researchers in the areas of loneliness and marital relationships but for clinicians as well, who would be able to identify specific shortcomings in the relationships which contribute to loneliness and thus be able to more specifically address them.

## 7. Limitations

There are several scales in use for assessing relationships, couple’s satisfaction, and loneliness. However, none addresses the specific issue of loneliness that may exist in intimate relations, which needs to be identified and measured. The present study was designed to integrate theoretical frameworks of loneliness in intimate relationships with empirical data, resulting in the development of a new scale, the Loneliness in Intimate Relationships Scale (LIRS). The preliminary versions of the LIRS were based on semi-structured questionnaires administered to a large heterogenic Israeli sample and on analyses of existing relevant scales. Exploratory and confirmatory factor analyses resulted in a well-structured 14-item scale depicting emotional, behavioral, and cognitive reactions and coping with loneliness in intimate relationships. The analyses revealed that the experience of loneliness in intimate relationships encompasses three main aspects: detachment, hurt, and guilt. Rokach and Brock [12] developed a scale to explore the qualitative aspects of loneliness in general. Their five-dimensional model included those very same factors: the hurt, which is a salient ingredient of loneliness, the hurt that follows feeling alone and unwanted, and the detachment that some people resort to in order to prevent further rejections, hurt, and pain. The LIRS evidenced excellent reliability levels with internal consistency estimates in the 90s. Furthermore, construct validity tests showed that, as expected, the LIRS subscales are positively related to higher levels of reported general and social loneliness and greater interpersonal needs and depression and are more prevalent among individuals who experienced loneliness in a relationship on a continuous, not state-related, basis.

Being connected is one of the most important human needs. It is so important that it affects not only emotional and physical well-being but is directly related to mortality rates. Studies have demonstrated a steep rise in mortality rates among socially isolated individuals [42]. However, being lonely does not necessarily reflect being unconnected with other human beings. The empirical and clinical literature conceptualizes loneliness as a two-dimensional construct, discriminating between objective loneliness, which reflects the quantity of one’s social interactions, and subjective loneliness, which concerns the quality of those interactions and reflects dissatisfaction with one’s social relationships, or as described by Weiss [43], as a “gnawing, chronic disease without redeeming features” (in [44], p. 446). Nevertheless, these two aspects of loneliness are not necessarily associated. According to Gottman’s [44,45] Distance and Isolation Cascade model, deterioration of marital distress can eventually result in disengagement, isolation, and loneliness. Gottman depicts the process leading from emotional flooding within a relationship to the feeling that any attempt to discuss problems will be pointless and futile—an approach that will eventually lead to emotionally parallel lives. When a couple reaches this advanced stage of relational deterioration, there is a complete absence of expressions of love and affection, and the partners experience emotional isolation, disengagement, and loneliness. Partners then find themselves emotionally uninvolved with and unavailable to each other. This “empty shell” marriage, which is characterized by partners’ disengagement and indifference, is a common antecedent of loneliness [46].

Marriage, similarly to any long-term intimate relationship, is an appropriate state for analyzing emotional loneliness separately from objective social isolation. The fact that loneliness in intimate relationships revealed itself not as a unitary concept but as being comprised of three stable and valid facets, namely, detachment, hurt, and guilt, may have important clinical implications, especially given the lack of couple-based therapy techniques designed specifically for the treatment of loneliness in close relationships [47]. As our studies revealed, detachment is a major factor of romantic loneliness, and it is thus reasonable to suggest that therapeutic interventions aimed at the improvement of attachment bonding and intimacy may potentially alleviate loneliness. More specifically, interventions that strengthen intimacy, emotional security, and mutual support are likely to encourage couples in therapy to reengage emotionally, thus reducing loneliness. Additionally, focusing on strengthening attachment bonding in romantic relationships is a preventive measure against relational loneliness. By employing preventive measures, educators and practitioners can utilize ways to help couples protect themselves from emotional detachment as a protective shield from romantic loneliness.

The authors note several limitations of these studies. First, the LIRS was not validated against behavioral measures. It is unclear how self-ratings on the LIRS manifest themselves behaviorally, thus limiting the scale’s ecological validity as a predictive tool for change. Future research should test the clinical utility of the LIRS to assess the level and type of relationship loneliness. Secondly, the LIRS was not examined in comparison to existing, general loneliness scales. While it is expected that the correlation would possibly be a moderate one, future research could examine it more closely. Thirdly, the interrelations between the three main subscales of the LIRS were not thoroughly examined. Different patterns of loneliness resulting from different combinations of levels of detachment, hurt, and guilt may manifest themselves in similar or different patterns of reactions to the stressful situation, thus calling for different types of psychological interventions. Finally, the LIRS was developed and validated with Israeli participants. This may, at present, limit the generalization of the application of this scale to other cultures. Future studies are needed in order to test the generalizability of the measurement model by testing it with other native English-speaking populations.

## Figures and Tables

**Figure 1 ijerph-19-12970-f001:**
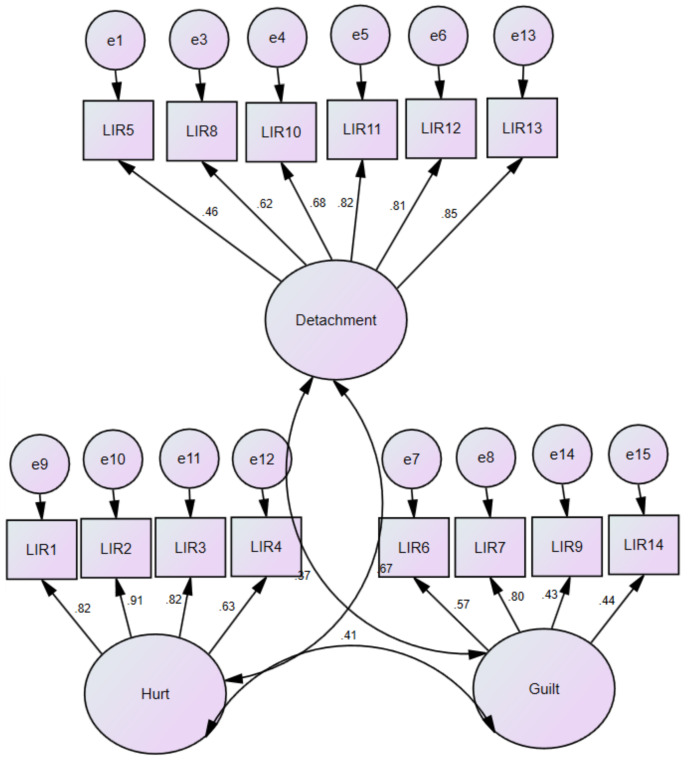
Standardized coefficients for the LIRS final version three-factor model—Study 3.

**Table 1 ijerph-19-12970-t001:** Stages of the development of the loneliness in intimate relationships scale for adults.

**Study 1**		Qualitative study aimed to generate items for the Loneliness in Intimate Relationships Scale (LIRS)
	Phase I	Generation of items based on a semi-structured open-ended questionnaire and removal of overlapping items
	Phase II	Domains identification and initial classification of items into categories
	Phase III	Collection of items derived from different sources: (1) theoretical and empirical literature relevant to loneliness in intimate relationships, and (2) impressions, evaluations and assessments derived from the clinical practice of two expert psychologists, specialists in couples therapy
	Phase IV	Item evaluation with respect to clarity, ambiguousness and comprehensibility
	Phase V	Item reformulation based on the item evaluations
**Study 2**		Final scale development (N = 215)
**Study 3**		Structural evaluation (N = 306)
**Study 4**		Construct validity

**Table 2 ijerph-19-12970-t002:** Categories derived from content analysis of the semi-structured questionnaires, relevant scales and clinical practice.

Category Name
Pain
Hopelessness
Emotional hurt
Anger
Frustration
Crying
Stress
Lacking partnership
Lack of intimacy
Misunderstanding
Lack of love
Sadness
Unavailability/lack of support
Emotional shut-off
Concerns about the fate of the relationship
Lack of appreciation
Helplessness
Insecurity
Self-blame
Hope for change
Low self-esteem
Perceived partner’s indifference to the relationship
Disengagement
Fear of end of relationship
Thoughts about suitability
Disrespect
Depression
Sense of failure
Self-pity

**Table 3 ijerph-19-12970-t003:** Factor loadings and statistics of the loneliness in intimate relationships second 14-item version—Study 3.

Item	Factor		
	Detachment	Hurt	Guilt
13. I thought about ending our relationship	** 0.78 **	0.29	0.08
12. I wondered whether we are suitable	** 0.74 **	0.24	0.20
8. I invested in the relationship without getting back	** 0.74 **	0.21	0−.08
11. I felt that I could not continue like this	** 0.73 **	0.35	0.15
10. I felt that I was not important to him/her	** 0.72 **	0.26	0.12
5. My husband/partner had no time for me	** 0.59 **	0.08	0.19
2. I felt hurt	0.34	** 0.86 **	0.06
1. I felt a deep sense of pain	0.24	** 0.83 **	0.17
3. I felt insulted	0.23	** 0.83 **	0.15
4. I was very tense	0.36	** 0.61 **	0.15
6. I felt that I was not fulfilling my part in the relationship	−0.02	−0.02	** 0.78 **
7. I felt guilty for my misdeeds in the marriage/relationship	0.13	0.24	** 0.74 **
14. I thought that I was infantile	0.06	0.18	** 0.61 **
9. I felt that I was not worthy of his love	0.24	0.03	** 0.56 **
Mean	3.41	3.96	2.62
SD	1.38	1.41	1.12
% Explained Variance	**40.5%**	**11.7%**	**8.7%**

Note: Bold and underlined loadings indicate the item’s affiliation to factors.

**Table 4 ijerph-19-12970-t004:** Goodness-of-fit of the confirmatory analyses of the original and random models of the english version of the LIRS (14 items)—Study 3.

Model	χ^2^	df	χ^2^ Difference	*p* Difference	IFI	CFI	RMSA	*p* RMSA
Original	246.44 ***	74			0.91	0.91	0.09	<0.001
Random	619.04 ***	74	372.60	<0.001	0.72	0.71	0.16	<0.001

Note: *** *p* < 0.001.

**Table 5 ijerph-19-12970-t005:** Descriptive statistics of the study measures—Study 4.

Scale	M	SD	95% CI
LIRS			
Detachment	3.44	1.37	3.29–3.59
Hurt	3.97	1.40	3.81–4.14
Guilt	2.62	1.12	2.49–2.76
UCLA	38.41	11.25	37.25–39.60
SElSA (Social)	2.64	1.25	2.51–2.79
INQ			
Perceived burdensomeness	1.58	1.13	1.45–1.72
Thwarted belongingness	3.46	0.77	3.38–3.54
BDI	11.14	9.86	8.69–11.05

**Table 6 ijerph-19-12970-t006:** Pearson correlations between the LIRS subscales and loneliness and depression measures.

	LIRS Subscales		
	Detachment	Hurt	Guilt
UCLA	0.20 ***	0.21 ***	0.34 ***
SELS-A (Social)	0.18 **	0.19 ***	0.27 ***
INQ			
Perceived burdensomeness	0.16	0.16	0.19 ***
Thwarted belongingness	0.24 ***	0.18 ***	0.24 ***
BDI	0.31 ***	0.31 ***	0.30 ***

Note: ** *p* < 0.003; *** *p* < 0.001.

**Table 7 ijerph-19-12970-t007:** Descriptive and Inferential Statistics of LIRS by Type of Loneliness.

	Type of Loneliness					
LIRS Subscales	Continuous		Specific		*F* (1, 280)	$\eta_{p}^{2}$
	M	SD	M	SD		
Detachment	4.04	1.27	3.30	1.33	19.09 ***	0.06
Hurt	4.44	1.24	3.89	1.37	9.87 ***	0.03
Guilt	3.00	1.23	2.46	1.03	14.70 ***	0.05

Note: *** *p* < 0.001.

**Table 8 ijerph-19-12970-t008:** Descriptive and Inferential Statistics of LIRS by Gender.

LIRS Subscales	Men		Women		*F* (1, 301)	$\eta_{p}^{2}$
	M	SD	M	SD		
Detachment	3.22	1.36	3.63	1.36	6.77 **	0.02
Hurt	3.65	1.38	4.24	1.38	13.55 ***	0.04
Guilt	2.76	1.19	2.50	1.06	4.16 *	0.01

Note: * *p* < 0.05, ** *p* < 0.01, *** *p* < 0.001.

## Data Availability

The raw data supporting the conclusions of this article will be made available by the authors without undue reservation.

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
