# Peer review of "Loneliness in Intimate Relationships Scale (LIRS): Development and Validation"

_ijerph, 2022, doi:10.3390/ijerph191912970_

Round 1

Reviewer 1 Report

In general I was impressed with this paper. I do think that a stronger case could be made for the need to develop a new measure of loneliness for couples, given the existence of scales to assess emotional loneliness and scales (not cited/used) to assess marital satisfaction. One would assume a high relationship between a marriage satisfaction scale and the LIRS.

I have several suggestions beyond the above:

1. given the age and gender composition of the samples, it would be prudent to examine gender and age effects in levels of LIRS as well as age/gender differences in its factor structure.

2. I wonder what bias is introduced when asking participants about loneliness "in an intimate relationship" vs. asking about loneliness "in your relationship with your partner or spouse"-clearly, persons might respond differently depending upon the personalization (or not) of the relationship in which loneliness is experienced. Likewise, responses might differ also between persons who were/were not currently in a steady relationship or were married vs. single vs. divorced (or widowed for that matter).

3. Can the authors report Kappas or correlations between judges as opposed to percentage of agreement as an index of interrater reliabilty?

4. On p. 4, "clarity and comprehensibility" should be more carefully defined.

5. On p. 7, the meaning of "mean eigenvalues and 95% eigenvalue obtained from from random data" need to be clarified. How was random data collected? What do the eigenvalues so used represent? IN comfirmatory factor analysis, it is common to compare a model where no common factors/no differential pattern of loadings exist to one which a priori specifies a given number of factors with a given pattern of factor loadings. Additionally, does this 3 factor model hold up across relationship status, gender, or age? Was duration of the relationship considered as an influence?

6. On p. 10, when stating patterns of convergent and discriminant validity, it would be better to a priori state what this pattern would be-this could also be tested via confirmatory FA. Again, the list of scales used excludes measures of marital satisfaction and perhaps measures of sadness, relationship satisfaction, communication clarity/satisfaction, or other measures indexing the quality of the relationship to one's partner or spouse.

7. On p. 12, could not the interrelations between the subscales be evaluated in the context of the confirmatory factor analyses?

Author Response

In general I was impressed with this paper. I do think that a stronger case could be made for the need to develop a new measure of loneliness for couples, given the existence of scales to assess emotional loneliness and scales (not cited/used) to assess marital satisfaction. One would assume a high relationship between a marriage satisfaction scale and the LIRS.

Thank you for the encouraging feedback about our manuscript.

I have several suggestions beyond the above:

  1. given the age and gender composition of the samples, it would be prudent to examine gender and age effects in levels of LIRS as well as age/gender differences in its factor structure.

We appreciate your pointing out the possibility of gender and age effects on LIRS subscales. We conducted analyses revealing interesting gender differences regarding the three LIRS subscales. The details of these are now described in Study 3. The LIRS subscales were not related to age, so these null results will not be presented.

For the possibility of gender and/or age effects on the LIRS factor structure, we agree that this might be important, both theoretically and practically, and we intend to explore these directions in our future research. Yet, we think that adding such analyses to the present research would be overwhelming and  a deviation from its main direction.

  1. I wonder what bias is introduced when asking participants about loneliness "in an intimate relationship" vs. asking about loneliness "in your relationship with your partner or spouse"-clearly, persons might respond differently depending upon the personalization (or not) of the relationship in which loneliness is experienced. Likewise, responses might differ also between persons who were/were not currently in a steady relationship or were married vs. single vs. divorced (or widowed for that matter).

You raised an interesting point. Since the scale focuses on ‘intimate relationships’ we thought that it would make a lot of sense to ask about loneliness specifically in intimate relationships. While it is possible, as you noted that having a partner while answering the questionnaire may have some effect on the responder, we suggest that asking -in our semi structured interview- “We would appreciate your reflections on an experience of loneliness in an intimate relationship “the participant will reflect on past experiences, either present or not where s/he felt lonely. We therefore, respectfully, doubt that whether the participant is in a relationship at present would make a significant difference.   In addition, descriptions based on  past relationships may provide a deeper insight into loneliness in intimate relationships.  

  1. Can the authors report Kappas or correlations between judges as opposed to percentage of agreement as an index of interrater reliabilty?

Thank you. We replaced percentage of agreement with Kappa at the final stage of item categorization.

  1. On p. 4, "clarity and comprehensibility" should be more carefully defined.
  2. On p. 7, the meaning of "mean eigenvalues and 95% eigenvalue obtained from from random data" need to be clarified. How was random data collected? What do the eigenvalues so used represent? IN comfirmatory factor analysis, it is common to compare a model where no common factors/no differential pattern of loadings exist to one which a priori specifies a given number of factors with a given pattern of factor loadings. Additionally, does this 3 factor model hold up across relationship status, gender, or age? Was duration of the relationship considered as an influence?

(a) Random data – We used O'Connor's (2000) software to generate eigenvalues (and 95% CI's) on the random data set. Each parallel data set is based on column-wise random shufflings of the values in the raw data matrix using Castellan's (1992, BRMIC, 24, 72-77) algorithm; The distributions of the original raw variables are exactly preserved in the shuffled versions used in the parallel analyses; Permutations of the raw data set are thus highly accurate and most relevant. 1000 datasets are   considered sufficient. We added O'Connor's (2000) reference when describing the parallel analysis.

(b) Regarding factor structure by gender, age, and status, please refer to our response to comment #1.

(c) Relationship duration was not considered as an influence. In the future, we plan to explore this important factor in more detail.

  1. On p. 10, when stating patterns of convergent and discriminant validity, it would be better to a priori state what this pattern would be-this could also be tested via confirmatory FA. Again, the list of scales used excludes measures of marital satisfaction and perhaps measures of sadness, relationship satisfaction, communication clarity/satisfaction, or other measures indexing the quality of the relationship to one's partner or spouse.

We mentioned that expectations in the introductory comments, and again in future directions for research.

  1. On p. 12, could not the interrelations between the subscales be evaluated in the context of the confirmatory factor analyses?

Thank you for your comment. We now refer to the interrelationships between the factors as emerged in the CFA. 

Reviewer 2 Report

Review of ijerph-1916227, Loneliness in Intimate Relationships Scale [LIRS]: Development and Validation

The world of surveys within psychology and public health research is full of scales with unclear background and validity. It is such a pleasure to read a manuscript in which the authors present a thorough and systematic development of a new scale. I have a couple of comments for the authors to consider.

The abstract is fine, but I recommend that the authors add the study aim in the abstract.

The introductory section entitled “Loneliness in Intimate Relationships Scale [LIRS]: Development and Validation” provides a nice and clear justification for the study aim: To describes research aimed at developing a new scale for assessing loneliness in relationships for adults and testing its basic psychometric properties regarding reliability and validity. The authors followed a systematic and standardized methodology for questionnaire construction which includes four studies, starting with an open-ended qualitative approach and ending with a study of the scale’s construct validity.

The presentation of applied methodology and findings in the five sub-studies is clear and straightforward.

Study 1 was a qualitative approach to generate items. Study 2 was a quantitative study to test scale properties of the first version of the new scale. The authors used factor analysis to study scale properties and removed inappropriate items, resulting in a new 14-item version of the new scale. Study 3 involved a study population of 306 volunteer participants to validate the structure pattern of the 14-item version of new scale and to analyze its psychometric properties. The study applied exploratory and confirmatory factor analyses resulted in identification of three factors: 1) thoughts and feelings concerning detachment and separation, 2) feelings of hurt and pain, and 3) sense of guilt and responsibility. In Study 4, the authors assessed the construct validity by comparison with other relevant scales, using a correlational approach.

The manuscript includes many tables, but they are justified and clear and easy to understand.

The general discussion presents a fine integration of the authors’ findings into the general literature about loneliness and marital relations. Further, the discussion suggest several ways to further test and develop the scale. In my opinion, the best way to assess construct validity of such a scale is by means of Rasch models, i. e. item response models. I think you learn much more about validity by using such models, for example you get specific information about possible differential item functioning. The authors may consider addressing this topic in the Discussion section.

Author Response

The world of surveys within psychology and public health research is full of scales with unclear background and validity. It is such a pleasure to read a manuscript in which the authors present a thorough and systematic development of a new scale. I have a couple of comments for the authors to consider.

Thank you very much for your encouraging comment.

The abstract is fine, but I recommend that the authors add the study aim in the abstract.

The following sentence was introduced in the abstract, addressing your comment: “Loneliness in intimate relationship may damage the relationship, if it goes on, and thus this newly developed scale has been introduced to add clinicians and researchers in discovering loneliness in an intimate union so it can be addressed before it negatively affects the union“.

The introductory section entitled “Loneliness in Intimate Relationships Scale [LIRS]: Development and Validation” provides a nice and clear justification for the study aim: To describes research aimed at developing a new scale for assessing loneliness in relationships for adults and testing its basic psychometric properties regarding reliability and validity. The authors followed a systematic and standardized methodology for questionnaire construction which includes four studies, starting with an open-ended qualitative approach and ending with a study of the scale’s construct validity.

The presentation of applied methodology and findings in the five sub-studies is clear and straightforward.

Study 1 was a qualitative approach to generate items. Study 2 was a quantitative study to test scale properties of the first version of the new scale. The authors used factor analysis to study scale properties and removed inappropriate items, resulting in a new 14-item version of the new scale. Study 3 involved a study population of 306 volunteer participants to validate the structure pattern of the 14-item version of new scale and to analyze its psychometric properties. The study applied exploratory and confirmatory factor analyses resulted in identification of three factors: 1) thoughts and feelings concerning detachment and separation, 2) feelings of hurt and pain, and 3) sense of guilt and responsibility. In Study 4, the authors assessed the construct validity by comparison with other relevant scales, using a correlational approach.

The manuscript includes many tables, but they are justified and clear and easy to understand.

The general discussion presents a fine integration of the authors’ findings into the general literature about loneliness and marital relations. Further, the discussion suggest several ways to further test and develop the scale. In my opinion, the best way to assess construct validity of such a scale is by means of Rasch models, i. e. item response models. I think you learn much more about validity by using such models, for example you get specific information about possible differential item functioning. The authors may consider addressing this topic in the Discussion section.

Dear reviewer, needless to say that reading your feedback, made us beam. Thank you for your encouraging words!!

Reviewer 3 Report

Loneliness in Intimate Relationships Scale [LIRS]: 2 Development and Validation

Review: Past measures of loneliness are agnostic to intimate relationships – some individuals may appear to have social support but the relationships don’t address the loneliness individuals may feel. The authors sought to address this research gap by creating the Loneliness in Intimate Relationships Scale (LIRS). The authors used four studies to generate items and conduct EFA and CFA analyses. The final scale consisted of 14 items.

The steps taken to develop the measure were extremely thorough and based on the suggested guidelines of the Standards for Educational and Psychological Testing. The paper’s major strength was using many samples to conduct each of the scale validation steps. I provide feedback below that I believe will help improve a strong paper.

Comments

·         Introduction: “although they nowdays have smaller chances of succeeding, they are still the preferred lifestyles”

o   Is this true? I thought that less people were marrying (choosing cohabitation) but that the divorce rate dropped for those that did marry? Also, I think would use another word than “nowdays” or rewrite the sentence.

·         General Comment: The authors included a few participants who were not in relationships at the time or did not report their relationship status. Although this is a small subset of the sample, could these participants be potentially biasing results?

·         Study 2: No feedback other than I think the authors did an extremely thorough job with the EFA analyses and description. (1) Oblique factor rotation is ideal -  I’ve reviewed other manuscripts where domains are theoretically correlated and they use the wrong rotation. They demonstrated that varimax (orthogonal) was ok to use; (2) I appreciate the participants-per-item ratio. One question – was principal axis or principal components analysis?

·         Study 2: If possible, can the authors add a bit more information about Parallel analysis, please? Also, where were the analyses performed (R or SAS, etc)?

·         Study 2: Based on the category list presented in Table 2, what items were poor loaders in study 2 or loaded on more than one category? If possible, can the authors list the loadings for all 66 items?

·         Study 3: conducting an additional EFA was very helpful, but I don’t think it was a good idea to perform both the EFA and the CFA on the same sample of 306 participants. Using the same sample for exploratory and confirmatory is problematic because the results aren’t necessarily being “confirmed” if they are coming from the same participants. A separate sample would need to be collected to confirm what was found with the exploratory sample. I suggest omitting the exploratory analyses and only presenting the confirmatory results and modifying Table 3 to be the results from Study 2 (see my previous point).

o   May also be useful to report the alphas of the scales that were extracted

·         Figure 1: For figure 1, can the authors move the latent correlations to be more legible, please? Also, I would add a sentence and/or a note in figure 1 to denote that all factor loadings and latent correlations were statistically significant.

·         Study 4. In the introduction, the authors mentioned relationship satisfaction scales and how these scales don’t capture loneliness in relationships. Were any relationship satisfaction scales administered to these participants?

o   Before running the correlations with the LIRS and the other scales, I would just reported the correlations between the three LIRS subscales.

Author Response

Loneliness in Intimate Relationships Scale [LIRS]: 2 Development and Validation

Review: Past measures of loneliness are agnostic to intimate relationships – some individuals may appear to have social support but the relationships don’t address the loneliness individuals may feel. The authors sought to address this research gap by creating the Loneliness in Intimate Relationships Scale (LIRS). The authors used four studies to generate items and conduct EFA and CFA analyses. The final scale consisted of 14 items.

The steps taken to develop the measure were extremely thorough and based on the suggested guidelines of the Standards for Educational and Psychological Testing. The paper’s major strength was using many samples to conduct each of the scale validation steps. I provide feedback below that I believe will help improve a strong paper.

Dear reviewer,

Thank you for your most positive feedback. It is encouraging to read it.

Comments

  • Introduction: “although they nowdays have smaller chances of succeeding, they are still the preferred lifestyles”
  • Is this true? I thought that less people were marrying (choosing cohabitation) but that the divorce rate dropped for those that did marry? Also, I think would use another word than “nowdays” or rewrite the sentence.

The research indicates that marriage is still the preferred way of cohabiting, so we left it as is, but did change the word ‘nowdays’ to ‘presently’.

  • General Comment: The authors included a few participants who were not in relationships at the time or did not report their relationship status. Although this is a small subset of the sample, could these participants be potentially biasing results?

Thanks for the question. It is possible, though not probable, that these few cases may bias the results, but we believe not in a significant way, if at all.

  • Study 2: No feedback other than I think the authors did an extremely thorough job with the EFA analyses and description. (1) Oblique factor rotation is ideal - I’ve reviewed other manuscripts where domains are theoretically correlated and they use the wrong rotation. They demonstrated that varimax (orthogonal) was ok to use; (2) I appreciate the participants-per-item ratio. One question – was principal axis or principal components analysis?

We applied principal components analysis.

  • Study 2: If possible, can the authors add a bit more information about Parallel analysis, please? Also, where were the analyses performed (R or SAS, etc)?

We added a detailed description of the parallel analysis which was conducted, as indicated, using  an SPSS Macro.

  • Study 2: Based on the category list presented in Table 2, what items were poor loaders in study 2 or loaded on more than one category? If possible, can the authors list the loadings for all 66 items?

The 66-item full early version of the questionnaire is now added as Appendix 1.

  • Study 3: conducting an additional EFA was very helpful, but I don’t think it was a good idea to perform both the EFA and the CFA on the same sample of 306 participants. Using the same sample for exploratory and confirmatory is problematic because the results aren’t necessarily being “confirmed” if they are coming from the same participants. A separate sample would need to be collected to confirm what was found with the exploratory sample. I suggest omitting the exploratory analyses and only presenting the confirmatory results and modifying Table 3 to be the results from Study 2 (see my previous point).

Your comment is greatly appreciated. We omitted the exploratory analysis from Study 3 and now describes the EFA in details in Study 2. The full EFA results of Study 2 are now presented in Table 3. 

o   May also be useful to report the alphas of the scales that were extracted

Alphas of the extracted scales are now added.

  • Figure 1: For figure 1, can the authors move the latent correlations to be more legible, please? Also, I would add a sentence and/or a note in figure 1 to denote that all factor loadings and latent correlations were statistically significant.

We moved the latent correlations to be clear and indicated in the figure that all coefficients are significant·

         Study 4. In the introduction, the authors mentioned relationship satisfaction scales and how these scales don’t capture loneliness in relationships. Were any relationship satisfaction scales administered to these participants?

The literature clearly established that loneliness is significantly and negatively correlated with relationship satisfaction, and so we did not see it necessary to duplicate that mode of research. In the future, we will, definitely, do so.

o   Before running the correlations with the LIRS and the other scales, I would just reported the correlations between the three LIRS subscales.

We now report the correlations between the LIRS subscales (Table 6).

Round 2

Reviewer 1 Report

The authors, for the most part, addressed most, but not all of my concerns. It is of publishable quality, however.